# Scaled-Model Radar Cross-Section Measurement: The Influence of the Scattered Field under Gaussian Beam Illumination—A Theoretical Analysis

**DOI:** 10.3390/s23167202

**Published:** 2023-08-16

**Authors:** Yi Xuan Ang, Hoi-Shun Lui

**Affiliations:** 1School of Electrical Engineering and Computer Science, The University of Queensland, St. Lucia, QLD 4072, Australia; 2School of Engineering, University of Tasmania, Sandy Bay, TAS 7005, Australia

**Keywords:** scaled-model radar measurement, radar cross-section, Gaussian beam, measurement by laser beam, scattering measurement

## Abstract

At microwave frequencies, radar cross-section (RCS) measurements are usually performed by placing the target in the far-field region of the antenna. The wavefront of the radiating field from the antenna can be approximated as planar, ensuring that the incident field and the power interact with the target independently of the antenna. However, for electrically large targets, the required distance becomes significant, posing challenges for implementation. Scaled-model RCS measurements offer an alternative solution. RCS measurements at terahertz and optical frequencies typically require a collimated beam as the source, where the intercepted power and RCS become dependent on the excitation. To address this dependency, researchers have proposed modifying the RCS definition to account for the intercepted power and to analytically formulate the scattering problem. However, such modifications require prior knowledge of the target’s geometry and material properties, which are often not readily available in practice. This also limits the study to only canonical targets. In this paper, we propose an alternative approach for modelling the intercepted power. The Gaussian beam is decomposed into a number of plane waves travelling to different directions using the theory of plane wave spectrum. The scattering problem is solved using the full-wave method of moment. Through theoretical proofs and numerical examples involving spheres and a non-canonical target, with a scaled-model aircraft, we demonstrate that the original RCS definition can serve as a good approximation for scaled measurements, provided that the beam waist is approximately four times the target’s dimensions. These findings provide valuable guidelines for radar engineers when performing scaled measurements using collimated beams. The results, which match those obtained from full-model measurements, enable us to predict the RCS of full-scale targets. This capability facilitates various target-related applications, such as target characterization, classification, detection, and even recognition.

## 1. Introduction

Target detection and tracking were the major applications upon a radar’s invention [1]. Over the last few decades, advances in digital signal processing and image processing have opened avenues for various sensing applications, such as radar imaging [2,3,4,5], automated target recognition from 1-D radar signals [6,7,8,9,10] and 2-D radar images [11]. Simultaneously, developments in metamaterials and frequency-selective surfaces have allowed for the better design of stealth objects [12,13] and surfaces with scattering reduction capabilities [14]. With the recent surge in the Internet of Things, 5G/6G communications, autonomous vehicles, and healthcare, there is a growing demand to understand the scattered strength from objects under the illumination of microwave radiation from various aspects and polarizations, i.e., radar cross-section (RCS) estimation.

Traditionally, RCS prediction can be accomplished through four major approaches: analytical prediction, computer simulation, full-scaled measurement, and scaled-model measurement [15]. For simple and canonical objects, such as spheres and plates, their RCS can be obtained analytically by solving Maxwell’s equations [13,16]. For geometrically complicated targets without available analytical results, full-wave and asymptotic computational electromagnetic techniques—such as the moment method, Physical Optics (PO), and the Geometrical Theory of Diffractions—allow for the accurate modelling of their structures. The scattered far field, and thus, the RCS of the target under plane wave excitation from any aspect and polarization can be numerically solved [15]. An experimentally, full-scaled measurement seems to be the most straightforward solution. In the transmitting end, the distance between the transmitting antenna and the target needs to be large enough to ensure that the electric and magnetic field components are orthogonal, resulting in a Transverse Electromagnetic (TEM) mode ($R_{a}>2D_{a}^{2}/\lambda$, where $D_{a}$ is the dimension of the transmitting antenna). This condition is crucial for most microwave RCS measurements, as it ensures that the illuminating wave interacting with the target is approximately uniform in space with a maximum phase error below π/8 [17]. The incident wave can be approximated as uniform, simplifying the analysis of the incident power density (see Equation (1) in Section 2). Furthermore, this measurement condition also aligns well with many military and health monitoring [18] applications, where the target is located far beyond $2D_{a}^{2}/\lambda$. In the receiving end, the receiving antenna needs to be in the range of $R_{t}>2D_{t}^{2}/\lambda$ [19] to measure the scattered far field, where $D_{t}$ is the dimension of the target. It is worth noting that both $R_{a}$ and $R_{t}$ are functions of wavelength. Although the potential of radar applications at resonant frequency ranges (0.4λ ≤ $D_{t}$ ≤ 10λ) has been studied [8,9,10], the majority of the work is in the optical region ($D_{t}$ >> 10λ), where the target is treated as a sum of point scatters, resulting in fairly large $R_{t}$ and $R_{a}$ in most cases. For example, measuring a 1 m target at 10 GHz (λ = 0.03 m, X-band) requires a range of 96 m, while measuring a 3 m target at 100 GHz (λ = 3 mm, W-band) requires a range of 6194 m [20]. Additionally, the measurement environment, whether indoor or outdoor, needs to be electrically quiet [15]. Facilitating RCS measurements in large-range, electrically quiet environments is not always trivial.

The scalability nature of Maxwell’s equations suggests that the same scattering measurement can be achieved by downscaling the target size (e.g., by a factor of 10) while increasing the illumination frequency by the same factor, provided that the target’s dimension in terms of wavelength remains the same and the material properties remain unchanged [21,22]. In 1978, a laboratory method for RCS measurement at 0.1 THz was reported and demonstrated using scaled ship models [23]. At terahertz and optical frequencies, scattering measurements are usually performed on optical tables using collimated beams from laser sources [3,13,24,25,26]. For imaging and sensing applications (e.g., terahertz microscopy), it is common practice to focus the collimated Gaussian beam to a small spot with maximum energy density using lenses and placing the target at the centre of this focal spot. Using the Gaussian beam model [27,28], the spot size of this focal spot can be approximated with the beam waist $w_{0}$. The transmitted and scattered signals are now collimated [3,13,26], rendering the aforementioned range conditions for both transmitting and receiving ends inapplicable. Independent studies conducted by different groups consistently show that the amplitudes of the RCS obtained using scaled-model targets and collimated beams are comparable but not identical to the ones obtained using physical optics approximations [24,25] and analytical solutions [13,16,29,30,31,32]. The discrepancy originates from partial illumination [25] and uneven illumination profiles in space [31,32] when a collimated beam is used.

The main objective of a scaled-model measurement is to provide a handy alternative to full-scale measurement at microwave frequencies, which can be challenging to facilitate due to the aforementioned range requirements. In designing stealth objects with specific RCS reduction requirements, a scaled-model measurement provides a cost-effective solution when it is expensive to fabricate full-scale prototypes [12,13]. Therefore, it is important to address the aforementioned discrepancy so that scaled-model measurements align well with the original problem.

To the best of our knowledge, the issue of non-uniform illumination has been reported but not explicitly handled in both theoretical and experimental contexts. In the theoretical study presented in [32], the team took into account the spatial variation of the Gaussian beam amplitude to calculate the RCS of the sphere. The results showed that the RCS varies with different beam waists. In the experimental context, the plane wave excitation condition was achieved in the experimental setup presented in [22], eliminating the need to consider this problem. In [13], the RCS is determined based on the reference power ($P_{ref}$) (see Section 4 of [13]), which is obtained from the emitted power ($P_{t}$), gain of the emitter antenna ($G_{t}$), effective area of the receiving antenna ($A_{r}$), and the distances between the target and the emitter ($R_{t}$) as well as between the target and the receiver ($R_{r}$), respectively. The actual power that interacts with the target is not explicitly included, as $P_{ref}$ (in [13]) is probably the best estimate, given $G_{t}$, $A_{r}$, $R_{t}$, and $R_{r}$, without prior knowledge of the target geometry. In our previous work [26], the reflectance (i.e., ${{|E}_{s}|}^{2}/{{|E}_{i}|}^{2}$) of a 3 mm metallic cube at 2.4 THz was measured using a 2″ collimated beam via laser-feedback interferometry (LFI), and it was found that the reflectance is proportional to the RCS. The LFI measurement is conceptually similar to a standard $S_{11}$ measurement using a vector network analyser, where the ratio of the reflected/incident power is captured. To the best of our knowledge, the discrepancy between the predicted RCS using a collimated beam and a uniform plane wave has not been properly resolved.

In this paper, we revisit this problem using full-wave moment method simulations [33]. In the next section, a brief review of RCS under plane-wave illumination is provided to enhance our understanding of the problem. Section 3 illustrates how RCS could be defined under Gaussian beam illumination. In Section 4, we first compare the scattered field and predicted RCS of a conducting sphere with a 10 mm diameter under both plane wave and Gaussian beam [29,31] illumination. Through a simple analysis, we identify a preferred configuration that minimizes the discrepancy. We then further validate this setting using a complex target—a scaled F5 aircraft model. Our results show that the preferred configuration provides a guideline to ensure that scaled-model measurements using Gaussian beams align with the RCS obtained under plane-wave illumination.

## 2. Radar Targets under Plane-Wave Illumination

Under uniform plane-wave illumination with the incident electric field, $E_{i}$, the power density of (W/m^2^) that illuminates the target can be expressed as follows [34]:(1)$$W_{i}=\frac{1}{2}\left({\left\lceil E_{i}\right\rceil }^{2}Y_{0}\right)$$

Here, $Y_{0}=1/\eta$ represents the intrinsic admittance in free space, and *η* represents the intrinsic impedance in free space. It is assumed that the target has an effective area, σ, commonly known as the radar cross-section (RCS). The power captured by the target can be given as follows:(2)$$P=W_{i}\times \sigma$$

We assume that the power captured by the target scatters evenly in all directions in three-dimensional space. The corresponding scattered power density at the distance *R* from the target is given using
(3)$$W_{s}=\frac{P}{4\pi R^{2}}=\frac{W_{i}\times \sigma }{4\pi R^{2}}=\frac{\sigma }{4\pi R^{2}}\frac{1}{2}\left({\left\lceil E_{i}\right\rceil }^{2}Y_{0}\right)$$

The scattered power density can also be written in terms of the scattered electric field as follows:(4)$$W_{s}=\frac{1}{2}\left({\left|E_{s}\right|}^{2}Y_{0}\right)$$

By equating these two equations, we obtain
(5)$$\frac{1}{2}\left({\left|E_{s}\right|}^{2}Y_{0}\right)=\frac{\sigma }{4\pi R^{2}}\frac{1}{2}\left({\left\lceil E_{i}\right\rceil }^{2}Y_{0}\right)$$

The RCS of a target under plane-wave illumination can be described by either of the following two expressions:(6)$$\sigma =4\pi R^{2}\frac{W_{s}}{W_{i}}=4\pi R^{2}\frac{{\left|E_{s}\right|}^{2}}{{\left|E_{i}\right|}^{2}}$$

## 3. RCS for Targets under Gaussian Beam Illuminations

For the lowest-order Gaussian beam (${TEM}_{0,0}$), the amplitude of the electric field distribution can be expressed as follows [27,28]:(7)$$\frac{E\left(x,y,z\right)}{E_{0}}=\left\{\frac{w_{0}}{w\left(z\right)}exp\left[-\frac{x^{2}+y^{2}}{w^{2}\left(z\right)}\right]\right\},\;\;\;\;where\;w^{2}\left(z\right)=w_{0}^{2}\left[1+{\left(\frac{z}{z_{0}}\right)}^{2}\right]\;\;and\;\;z_{0}=\frac{\pi nw_{0}^{2}}{\lambda_{0}}.$$

Here, *n* is the refractive index of the media, and $w\left(z\right)$ represents the beam width at position *z*. The beam waist, $w_{0}$, specifies the “width” of the beam along the orthogonal plane (*x* and *y* directions), and $z_{0}$ is the position along the *z*-axis where the beam width $w\left(z=z_{0}\right)$ becomes $w_{0}\sqrt{2}$ and the amplitude of the electric field becomes $1/\sqrt{2}$ of $E_{0}$. The distance from z=0 and $z=z_{0}$ is known as the Rayleigh range, which specifies the “depth” of the beam along the direction of propagation *z*. Figure 1a illustrates the relationship between these parameters. The red line outlines the beam width along the *z*-direction, where the amplitude of the electric field is down by 1/e of its peak value at $x^{2}+y^{2}=0$, i.e., the 1/*e* point [27].

Figure 1b,c show the cross-sectional view (*y*-*z* plane) of the electric field distribution at 1 THz ($\lambda_{0}=0.1\;mm$) with $w_{0}$ equal to 10 mm and 40 mm, respectively. These plots cover the same area of ±0.1 m and ±18 m along the y and z directions, respectively. It is evident that the field distribution of the two beams differs as we increase $w_{0}$ from 10 mm to 40 mm. The x-y cut of the latter beam at z=0 and $z=z_{0}$ is shown in Figure 1d,e, indicating that the beam width varies along *z*.

In this work, we assume the target is located at the centre of the beam along the beam waist $\left(x=0,y=0,z=0\right)$. With this assumption, $W_{i,GB}\left(E_{0},w_{0}\right)$ is now a function of both the incident field amplitude $E_{0}$ and the beam waist $w_{0}$. Similar to Equation (2), the corresponding power captured by the target is given using
(8)$$P_{i,GB}={W_{i,GB}\left(E_{0},w_{0}\right)\times \sigma }_{GB}$$
and the corresponding scattered power density by the target is given using
(9)$$W_{s,GB}=\frac{P_{i,GB}}{4\pi R^{2}}=\frac{{W_{i,GB}\left(E_{0},w_{0}\right)\times \sigma }_{GB}}{4\pi R^{2}}.$$

Once again, the scattered power density can also be written in terms of the scattered electric field as
(10)$$W_{s,GB}=\frac{1}{2}\left({\left|E_{s,GB}\right|}^{2}Y_{0}\right)$$

Equating Equations (9) and (10) gives
(11)$$\sigma_{GB}=4\pi R^{2}\frac{\frac{1}{2}\left({\left|E_{S,GB}\right|}^{2}Y_{0}\right)}{W_{i,GB}\left(E_{0},w_{0}\right)}$$

Equation (11) provides the corresponding RCS of a target under Gaussian Beam illumination. 

The question now is how to handle $W_{i,GB}\left(E_{0},w_{0}\right)$—the power density incident on the target, which is still a function of $w_{0}$. In [31], the incident power density of a sphere is analytically computed given the radius of the sphere. However, such calculations can only be made if the target geometry is known. For non-canonical targets and measurements where the target is physically provided, computing the integral can be practically challenging.

Similar to the approach reported in [13], we need to find a way to approximate the incident power density, $W_{i,GB}\left(E_{0},w_{0}\right)$. Here, we consider the total power $P_{0}$ passing through a circle of radius *r* in the transverse plane at position *z* (i.e., *r* is measured from (*x* = 0, *y* = 0, *z* = *z*′). At different positions along *z* for example, at z=0 and $z=z_{0}$ shown in Figure 1d,e, although the electric field distributions are not the same, the total power on any transverse plane (*z* = *z*′) equals the total power $P_{0}$ [27] due to the conservation of energy. The radius *r* measured from (*x* = 0, *y* = 0, *z* = *z*′) determines the amount of power captured within the circle. Mathematically, this can be written as [27,28]
(12)$$P\left(r,z\right)=P_{0}\left[1-e^{-2r^{2}/w^{2}(z)}\right]$$
(13)$$\mathrm{where}\;P_{0}=\frac{1}{2}\frac{E_{0}^{2}}{\eta }\left(\frac{\pi \omega_{0}^{2}}{2}\right)=\frac{E_{0}^{2}\pi \omega_{0}^{2}Y_{0}}{4}.$$

We make use of this as a model to approximate the amount of power captured by the target. Using Equation (12) as our estimate for $P_{i,GB}$ and substituting it in Equation (11), the “average power density” can thus be given using [27,28]
(14)$$W_{i,GB}=\frac{P\left(r,z=0\right)}{\pi r^{2}}=\frac{E_{0}^{2}w_{0}^{2}Y_{0}}{4r^{2}}\left[1-exp\left(-\frac{2r^{2}}{w_{0}^{2}}\right)\right].$$

Equation (14) represents the power density of the beam that passes through the circular area with radius *r*. We are aware that the power captured within the circular region is not closely related to the physical problem, i.e., the power captured by the target. However, this approximation provides us with a simple mathematical model that relates the beam waist and the physical dimension of the beam for estimating $P_{i,GB}$ in (8) and subsequently $W_{i,GB}$ in (11). It also enables us to visualize the size of the circle with respect to the beam waist. Here, we use this as an initial assumption to estimate the incident power density on the target; $W_{i,GB}$ as $w_{0}$ is known, and *r* can easily be determined. We then evaluate the effectiveness of this estimation through numerical results.

Given that the beam waist, $w_{0}$, is usually known (which can be measured experimentally [35,36]), the next question is to identify what *r* should be. It would be sensible to begin our estimation by setting *r* to be at least the largest dimension of the target, ensuring that the target is fully covered. In the next section, we evaluate this assumption using different values of *r* together with full-wave electromagnetic simulation.

## 4. Results 

To better understand how we can obtain meaningful predicted RCS data using scaled-model measurements via collimated beam illumination that matches well with the original full-scale measurements under plane-wave illumination, we examine the bistatic scattering of a conducting sphere and a scaled-model F5 aircraft. Using commercial full-wave moment method solver FEKO [33], we calculate the scattered electric field and RCS of the targets under plane-wave illumination, denoted as $E_{s,PW}$ and ${RCS}_{s,PW}$, respectively, using the moment method. Next, we calculate the scattered far field and RCS under Gaussian beam illumination, denoted as $E_{s,GB}$ and ${RCS}_{GB}$. The calculation of the scattered field under Gaussian beam illumination is not directly available in FEKO. However, using the theory of the plane-wave spectrum, we can decompose the field pattern under Gaussian beam illumination, which is a function of the wavelength (λ) and beam waist ($w_{0}$), as a sum of a number of plane waves with different incident angles [37]. By expressing the Gaussian beam illumination as a sum of plane-wave excitations, we can solve the electromagnetic scattering problems under Gaussian beam illumination. This allows us to obtain the scattered far field $E_{s,GB}$ towards different directions under the Gaussian beam illumination with beam waist $w_{0}$. To evaluate the impact of collimated beam illumination, we compare the $E_{s,GB}$ obtained using Gaussian beams with different beam waists to the scattered field obtained using plane-wave excitation ($E_{s,PW}$). Additionally, to obtain the RCS under Gaussian beam illumination, $\sigma_{GB}$, in Equation (11), we calculate the incident power density, $W_{i,GB}$, in Equation (14) with different radii, *r*, and use it as an approximation of the incident power density to calculate ${RCS}_{GB}$ in Equation (11). We compare the obtained $\sigma_{GB}$ under different values of *r* to the RCS of the same target under plane-wave excitation.

Let us now move on to the specific examples.

### 4.1. Sphere

First, we consider the bistatic configuration of a 10 mm conducting sphere target, as shown in Figure 2. In our examples, we consider the same target under both plane-wave and Gaussian beam illumination. Figure 2 illustrates the incident plane wave propagating along the negative z direction with the electric field component linearly polarized along the negative x direction.

To model Gaussian beam illumination, we utilize the theory of plane-wave spectrum. The Gaussian beam is represented using a total of 1891 (61 × 31) plane waves, encompassing 61 different angles along θ and 31 different angles along ϕ. In this context, θ=0° corresponds to the z-axis, and the range of θ is determined with the beam waist, $w_{0}$. The values of ϕ cover the range of [−90°, 90°]. The target is illuminated using a Gaussian beam with different beam waists, $w_{0}$. The corresponding bistatic scattered field $\left|E_{s,GB}\right|$ is shown in Figure 3a,b, compared to the scattered field $\left|E_{s}\right|$ obtained using plane waves. It is evident that when $w_{0}$ is 10 mm, $\left|E_{s,GB}\right|$ deviates from $\left|E_{s,PW}\right|$ as the bistatic angle exceeds 60 degrees. However, as the beam waist is increased to 40 mm and above, $\left|E_{s,GB}\right|$ becomes practically identical to $\left|E_{s,PW}\right|$.

To quantify the differences between $\left|E_{s,GB}\right|$ and $\left|E_{s,PW}\right|$, we calculate the mean-squared error (MSE) for each $w_{0}$. The corresponding results are listed in Table 1. The error decreases as the beam waist is increased from 10 mm to 40 mm. However, the error increases again when the beam waist is further increased. This suggests that the optimal beam waist in this case is 40 mm, which is four times greater than the diameter of the sphere. Thus, we establish the following prediction rule for estimating the optimal beam waist:(15)$$\mathrm{Optimum}w_{0}=4\times \mathrm{Maximum}\;\mathrm{dimension}\;\mathrm{exposed}\;\mathrm{to}\;\mathrm{radar}$$

Figure 4 illustrates the bistatic RCS, $\sigma_{GB}$, of the sphere. These values are computed by considering different radii, *r*, in Equation (14) to estimate the incident power density. The beam waist is fixed at the optimal value ($w_{0}=40\;\mathrm{mm}$) and the corresponding scattered electric field $E_{s,GB}$ is used in the calculation. The results are compared to $\sigma_{PW}$. Surprisingly, the results show that $\sigma_{GB}$ approaches $\sigma_{PW}$ as we reduce *r* from ${5w}_{0}$ down to $0.1w_{0}$. The MSE between $\sigma_{GB}$ and $\sigma_{PW}$ is listed in Table 2. The MSE decreases as the radius is decreased. These results suggest that a smaller radius is more desirable. Furthermore, the optimal choice of *r* is not dependent on the geometry of the target.

To delve deeper, we evaluate the limit of the incident power density as r approaches 0. Using L’Hôpital’s rule (shown in the Appendix A),
(16)$$\underset{r\to 0}{\lim}W_{i,GB}=\frac{{\left|E_{0}\right|}^{2}Y_{0}}{2}$$

The RCS of the target can be simplified to
(17)$$\sigma_{GB}=4\pi R^{2}\frac{{\left|E_{s,GB}\right|}^{2}Y_{0}}{2W_{i,GB}}=4\pi R^{2}\frac{{\left|E_{s,GB}\right|}^{2}}{{\left|E_{0}\right|}^{2}}$$

When the beam waist is optimal, such that $\left|E_{s,GB}\right|\approx \left|E_{s}\right|$:(18)$$\sigma_{GB}=4\pi R^{2}\frac{{\left|E_{s,GB}\right|}^{2}}{{\left|E_{0}\right|}^{2}}\approx 4\pi R^{2}\frac{{\left|E_{s}\right|}^{2}}{{\left|E_{i}\right|}^{2}}$$

The right-hand side of Equation (18) is identical to the RCS of the target under plane-wave illumination. In summary, if the target is illuminated using a Gaussian beam with a beam waist such that the scattered field is well matched with the scattered field obtained under planewave illumination, the obtained RCS will be comparable to the RCS under plane-wave illumination.

### 4.2. Scaled Model Aircraft (From the Front)

Based on our findings from the sphere example, we apply the prediction rule (Equation (15)) to estimate the optimal beam waist for a scaled-model F5 aircraft. The geometry of the F5 aircraft and the corresponding bistatic RCS configuration with the incident wave coming from the front of the aircraft are shown in Figure 5a,b. The model is made of a perfectly electric conductor (PEC). The corresponding lengths, $L_{1}$ and $L_{2}$, of the original problem at 1 GHz and the scaled-model aircraft at 1 THz are listed in Table 3. The entire geometry is reduced by a factor of 1000 as the frequency is increased by a factor of 1000. Using the rule derived from the sphere example, the predicted optimum beam waist is expected to be 32.8 mm (4 × 8.2 mm). Figure 5c shows the $\sigma_{GB}$ of the scaled model aircraft when illuminated using a Gaussian beam with the predicted optimum beam waist. In the calculation of the incident power density, *r* is selected to be 0.1$w_{0}$. The result is compared to $\sigma_{PW}$, and the MSE of $\sigma_{GB}$ with respect to $\sigma_{PW}$ is computed to be $5.3724\times 10^{-12}$. The error is negligible compared to the magnitude of RCS (above $10^{-3}$), which implies that the predicted RCS under Gaussian beam illumination matches well with the RCS under plane-wave illumination.

As a sanity check, Figure 6 shows the simulated scattered field $\left|E_{s,GB}\right|$ of the aircraft when illuminated with a Gaussian beam with different beam waists $w_{0}$. The results are compared to $\left|E_{s,PW}\right|$. The MSE between $\left|E_{s,GB}\right|$ and $\left|E_{s,PW}\right|$ of the aircraft is tabulated in Table 4. It is observed that the error decreases when the beam waist is increased from 10 mm to 30 mm and increases again when the beam waist is further increased. Therefore, the optimum beam waist obtained from the simulation results is 30 mm. Comparing the predicted optimum beam waist with the simulated value, the prediction rule gives a promising result and estimates the optimum beam waist with an accuracy of approximately 91%.

### 4.3. Scaled Model Aircraft (From the Top)

In the last example, the same scaled-model F5 aircraft is considered. The orientation of the aircraft is rotated such that the incident beam is illuminating the top of the aircraft, as shown in Figure 7a. Using $L_{2}$ as the reference geometry, the predicted optimal beam waist is now 56.8 mm. The aircraft is now illuminated using a Gaussian beam with the optimal beam waist. The corresponding RCS is calculated in the same manner. As shown in Figure 7b, it is clear that the results match well with the RCS obtained under plane-wave excitation. The MSE of the two RCSs is found to be $1.3635\times 10^{-7}$, which is practically negligible given that the RCS is at the level of $10^{-3}$ and above. The corresponding bistatic scattered electric field $\left|E_{s,GB}\right|$ is shown in Figure 8. The MSE of $\left|E_{s,GB}\right|$ and $\left|E_{s,PW}\right|$ is computed and listed in Table 5. It is deduced that the optimal beam waist is 50 mm, where the minimum value of MSE is found. The percentage error is found to be 13.6%.

## 5. Discussions 

### 5.1. Discussions of the Results 

The scattered electric field and radar cross-section (RCS) obtained using plane-wave excitation and Gaussian beam illumination were studied using the full-wave moment method. Using the scattered field under plane-wave illumination as our reference, we compared it with the scattered electric field obtained from Gaussian beam illumination by computing the mean-squared error (MSE). For a 10 mm diameter sphere target, we found that the minimum MSE value occurs when the beam waist is 40 mm, which is four times the largest dimension of the target. Using this as a guideline, we estimated the optimal beam waist required for a scaled-model F5 aircraft model. Our results show that this guideline allows us to obtain small errors that are comparable to the minimum case.

The calculation of the RCS requires knowledge of the incident power density on the target. For Gaussian beam illumination, the incident power density is complicated as the amplitude and phase of the electric field vary spatially. In this paper, we introduced an approximation to study how the RCS varies with different estimations of the power density. Through full-wave simulation and mathematical analysis, we found that the incident power density of plane-wave illumination with an electric field amplitude of $\left|E_{0}\right|$ is a good approximation when the beam waist of the Gaussian beam used in the RCS measurement is around four times the largest dimension of the target.

The findings in this paper can be understood from a different perspective. When the beam waist is large enough, which we found has to be at least four times the largest dimension of the target, the beam diameter (2 × beam waist) is eight times the target’s size. In other words, the target covers the region within $w_{0}/8$. The amplitude of the electric field within this region is very close to $\left|E_{0}\right|$. Therefore, the scattered field |$E_{s,GB}$| matches well with $|E_{s,PW}$|. Under such conditions, the approximation of using the incident power density from a plane wave, $W_{i}$ in Equation (1), becomes a very good approximation to the problem.

### 5.2. Previous Studies Reported in the Literature

The electromagnetic scattering of conducting cylinders, semicircular bosses, and spheres under the illumination of Gaussian beams has been studied in previous works [29,30,31,32]. In those studies, the Gaussian beam was positioned at a distance away from the target to simulate a beam emitted from a source, and the beam width was set to match the width of the source. However, in our current paper, we adopt a different approach by modeling the incident field distribution surrounding the target as a Gaussian beam profile. In practical scenarios, the size of this distribution, represented by the beam waist in our study, can be adjusted through collimation using lenses or mirrors with different focal lengths. Hence, there exists a distinction between the two problem formulations, and they are not strictly identical.

Regarding the conducting sphere case, previous results [31] indicate that when the beam width of the Gaussian beam is set to two times the diameter (four times the radius) of the sphere, the monostatic RCS error remains below 0.5 dB. Moreover, increasing the beam width further to four times the diameter (eight times the radius) of the sphere brings the results closer to those obtained under plane wave excitation. These observations align well with the findings of our moment method simulation results.

In summary, our study highlights the flexibility of modeling incident field distributions using Gaussian beam profiles and the adjustability of the beam waist through appropriate collimation. The comparison with previous works demonstrates the significance of selecting an appropriate beam width, which affects the accuracy of the monostatic RCS calculations. Such insights provide valuable guidelines for optimizing Gaussian beam-based simulations in electromagnetic scattering studies.

### 5.3. Scaling in Electromagnetic Scattering 

The present study is based on scaled-model testing, where we investigate the electromagnetic scattering of a F5 aircraft at 1 GHz (L-band) by applying a scaling factor *K* = 1000. This involves increasing the operating frequency from 1 GHz to 1 THz by multiplying *K*, while simultaneously reducing the dimensions of the target by multiplying 1/*K*. 

In addition, the resultant RCS obtained from scaled-model measurements needs to be multiplied by $K^{2}$ in linear scale or, equivalently, ${10log}_{10}(K^{2})$ in dBsm scale. For example, if the RCS of a scaled target with the largest dimension of 1 mm at 1 THz is found to be $10^{-3}\;m^{2}$ in linear scale or, equivalently, −30 dBsm in dBsm scale. The full-scaled target, which is 1000 times larger (i.e., 1 m), would have an RCS of $10^{3}\;m^{2}$ at 1 GHz, or equivalently 30 dBsm.

It is worth noting that different scaling factors could be used as long as the size of the target (e.g., $L_{1}$ and $L_{2}$ dimensions of the aircraft) and the illuminating wavelength (λ) maintain the same ratio, provided the material properties remain constant. However, it is challenging to have exactly the same material properties across the entire frequency spectrum, as material characteristics often vary with frequency. Identifying material with same material properties at a different frequency is a very challenging problem. Given these limitations, typical scaling factors fall within the range of 10:1 to 60:1. However, it is worth noting that a scaled measurement of 150:1 is documented in [22], and a reported scaled measurement of 250:1 can be found in [13]. The scaling of 1000:1 employed in our current study may likely become feasible through experimental realization in the near future. Further discussion on the scaling of target size, illumination frequency, and the challenges associated with matching material properties can be found in [38,39].

Recently, Perälä et al. [40,41,42,43] proposed a novel RCS prediction method that does not require a priori knowledge of the target’s material properties. In this method, the target’s geometry is extracted from optical images captured using a camera. The scattering centres of the target, as observed under visible light illumination, are then used as the basis for estimating the RCS at microwave frequencies. Although this approach assumes similarities in the propagation of microwave radiation and optical waves, we acknowledge that diffraction dominates in many microwave scenarios. The method applies the well-known expression $\sigma =4\pi A^{2}/\lambda^{2}$, which represents the RCS of a conducting plate with area A, to model the RCS of individual scattering centres. This assumption further considers the scattering centres to be perfectly electrically conducting (PEC), resembling the traditional point scatterer model for RCS prediction. As a result, non-metallic structures such as tires and glass windows may not be accurately modelled. The feasibility of this approach is demonstrated through a comparison of the measured RCS of a Volkswagen Golf car at S-band (2 GHz to 4 GHz) and reconstructed Inverse Synthetic Aperture Radar (ISAR) images.

Given that the length of the car is approximately $L_{golf}$ = 4 m and $\lambda_{2\mathrm{GHz}}=0.15\;m$, resulting in $L_{golf}/\lambda_{2\mathrm{GHz}}$ = 26, the size of the target is relatively large compared to the wavelength, leading to high-frequency phenomena dominating the electromagnetic scattering process. For high-frequency scattering phenomena, the primary focus lies in the existence, spatial position and types of scattering phenomena, and the material properties can be treated with some flexibility. However, if high precision is a strict requirement, applying coating material on the surface of scaled models could be considered [44]. Detailed information on the design of optical coatings can be found in [45].

### 5.4. About Measurement Environment

In RCS measurements conducted at radio frequencies, the establishment of a controlled and shielded testing area is paramount to minimize electromagnetic interference and ambient noise levels. Such an environment is especially critical when dealing with low radar cross-section targets, which demand high sensitivity and precision. Typically, RCS measurements are performed in anechoic chambers or Faraday cages, effectively shielding external electromagnetic interferences arising from wireless communication, electrical machinery, and electronic devices.

In the context of scaled-model RCS measurements, where targets are measured at sub-terahertz frequencies and above, multiple approaches are available, including electrical and optical methods. For instance, when lasers are employed as the source, measurements are commonly performed on optical tables [3,13,21,24,25,26,46] where the impact of electromagnetic interference becomes less significant. Nevertheless, calibration procedures play a crucial role in establishing the baseline response of the measurement system and accounting for any system biases or errors.

Furthermore, the test range itself is meticulously designed to minimize reflections and scattering caused by surrounding objects and clutter. By carefully controlling the measurement environment and employing proper calibration techniques, accurate and reliable RCS measurements can be achieved, enabling an in-depth analysis and characterization of targets at various frequency ranges. An example of a scaled model measurement system at 0.585 THz, utilizing a continuous-wave laser, can be found in [46].

### 5.5. Summary

Scale-model testing has garnered significant interest due to the substantial cost reduction in building and testing models compared to full-scale targets. When conducting scaled-model measurements at Terahertz frequencies, optical tables and laser sources with collimation are necessary. Previous studies have highlighted the dependence of measured RCS on the beam width under Gaussian beam illumination [31,32], especially when the beam width is small, resulting in partial target illumination [25]. In this study, we adopt a different approach, analysing the size of the beam that illuminates the target using the full-wave electromagnetic modelling and the method of moment. The moment method surpasses the constraints of analytical studies [31,32], which were limited to canonical targets only. Our findings demonstrate that the RCS obtained using Gaussian beams approaches those obtained using plane waves when the beam waist is at least four times the largest dimension of the target. This insight provides a critical condition for determining the appropriate beam size in scaled-model measurements. Ultimately, the derived scaled-model RCS enables us to predict the RCS of full-scale targets, facilitating target characterization, classification, detection, and even recognition applications.

## 6. Conclusions

In this paper, we studied the discrepancy of the radar cross-section (RCS) obtained using collimated beam and plane-wave excitations. Our results demonstrate that when the target is positioned at the centre of the focus and the beam waist is at least four times the largest dimension of the target (equivalently, the diameter of the beam is eight times the largest dimension of the target), the electric field interacting with the target becomes fairly homogeneous with an amplitude close to ${|E}_{0}$|. As a result, the incident power density commonly used for traditional RCS calculations under plane-wave excitation can be applied. Through full-wave moment method simulations, we also observed that the amplitude of the scattered field matches well with the one obtained under plane-wave illumination, and consequently, the resultant RCS. These findings address the concerns regarding partial illumination and the associated errors when measuring RCS using collimated beams [24,25,29,30,31,32]. They provide valuable insights for engineers involved in designing scaled-model RCS measurements using collimated beams.

## Figures and Tables

**Figure 1 sensors-23-07202-f001:**
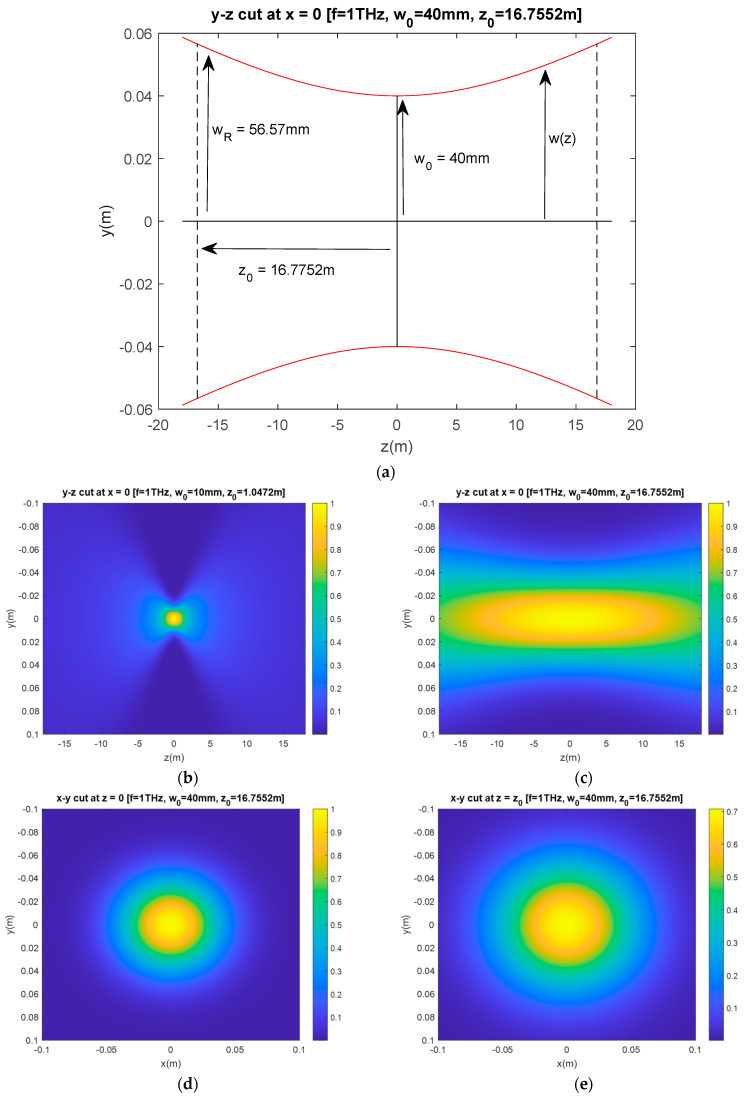
Examples of Gaussian Beam profile (TEM_0,0_) at 1 THz. (**a**) This figure illustrates the relationship between beam width *w*(*z*), beam waist, *w*_0_ and Rayleigh range z_0_. In this example, *w*_0_ = 40 mm. (**b**–**e**) are cross sectional views of the normalized electric field distribution of Gaussian Beam propagating along the *z* direction. The beam waists are (**b**) w_0_ = 10 mm (*y*-*z* cut at *x* = 0) and (**c**) w_0_ = 40 mm (*y*-*z* cut at *x* = 0), as well as (**d**) w_0_ = 40 mm (*x*-*y* cut at *z* = 0) and (**e**) w_0_ = 40 mm (*x*-*y* cut at *z* = *z*_0_), respectively.

**Figure 2 sensors-23-07202-f002:**
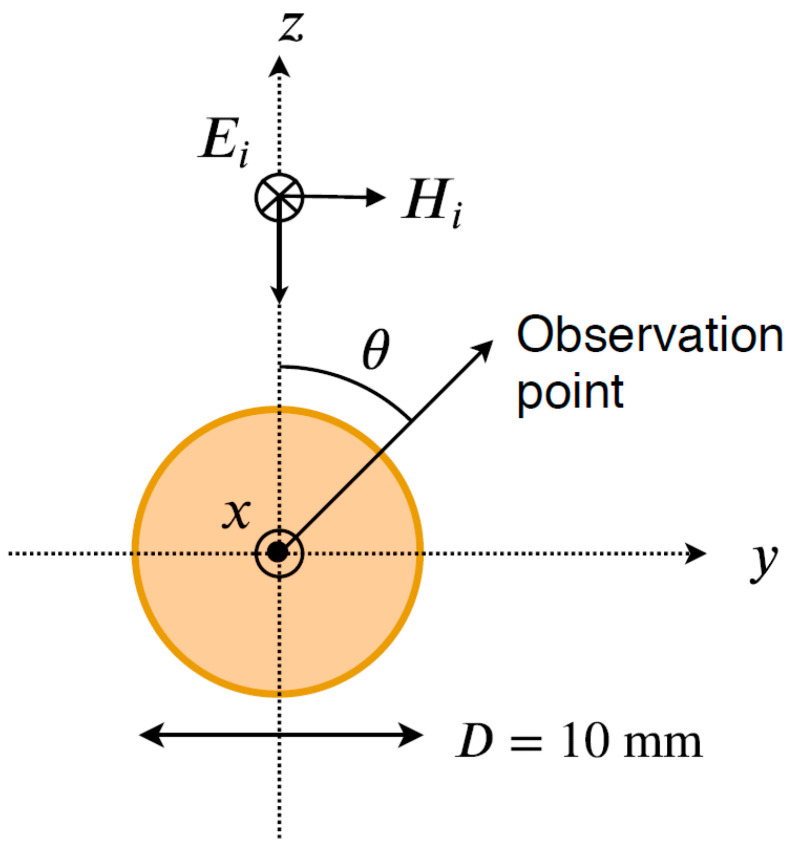
Bistatic configurations of a 10 mm diameter PEC sphere.

**Figure 3 sensors-23-07202-f003:**
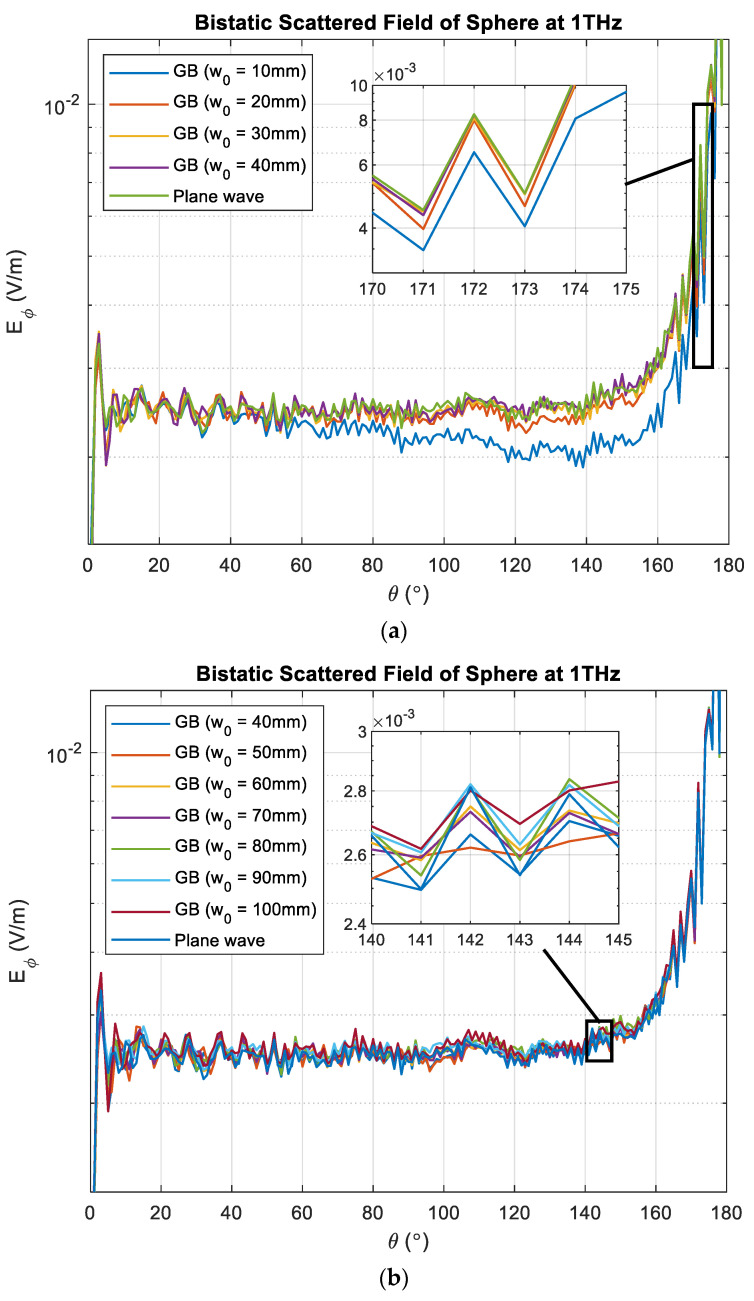
Bistatic scattered field of a 10 mm diameter PEC sphere. The sphere is illuminated using plane wave and Gaussian beam with (**a**) $w_{0}$ = 10 mm to 40 mm and (**b**) $w_{0}$ = 40 mm to 100 mm.

**Figure 4 sensors-23-07202-f004:**
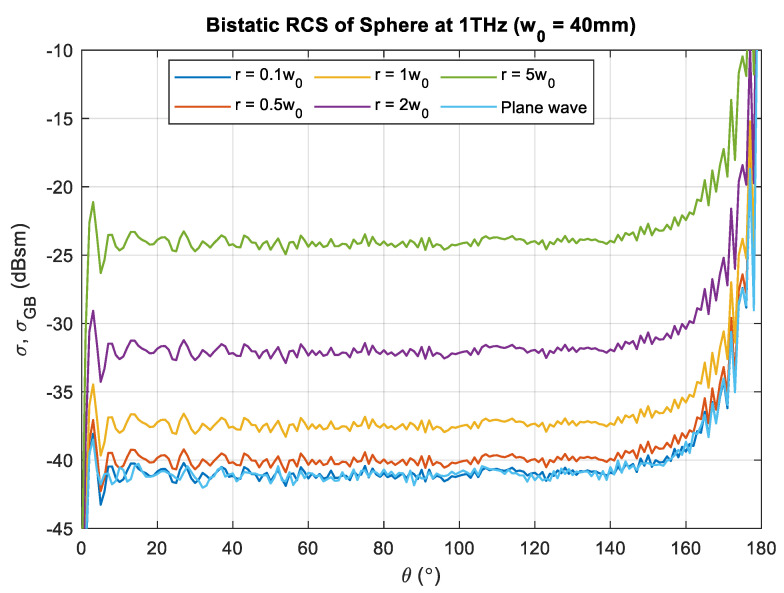
Bistatic RCS of the 10 mm sphere plotted against scattered angle. The sphere is illuminated using a Gaussian Beam with *w*_0_ = 40 mm. The RCS is calculated using Equations (11) and (14).

**Figure 5 sensors-23-07202-f005:**
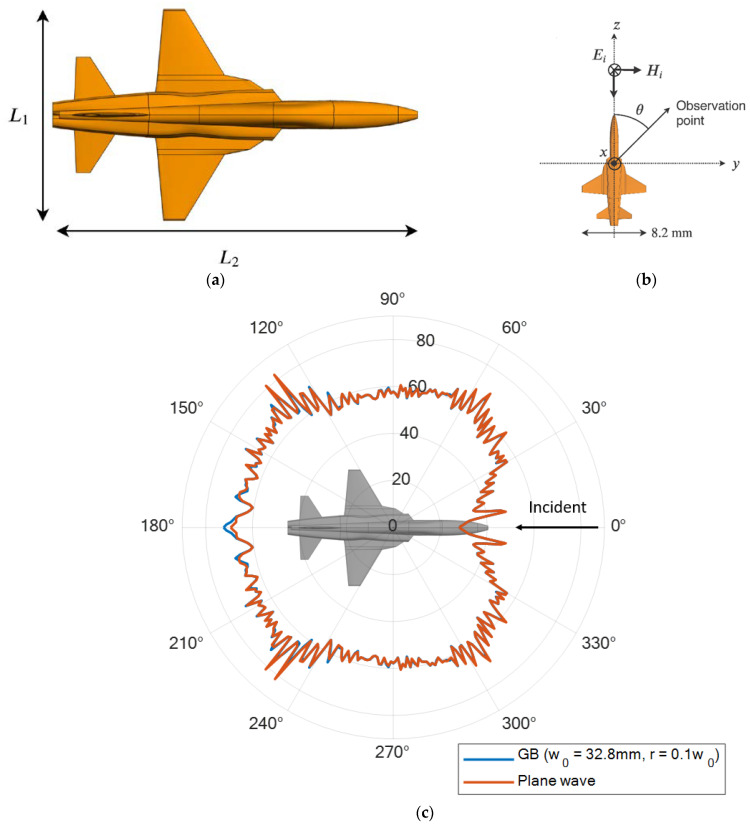
(**a**) Geometry of the F5 aircraft (left) and (**b**) the bistatic RCS configuration of the F5 aircraft with Gaussian beam illumination coming from the front (right). (**c**) Bistatic RCS result obtained from the predicted “optimal” beam waist.

**Figure 6 sensors-23-07202-f006:**
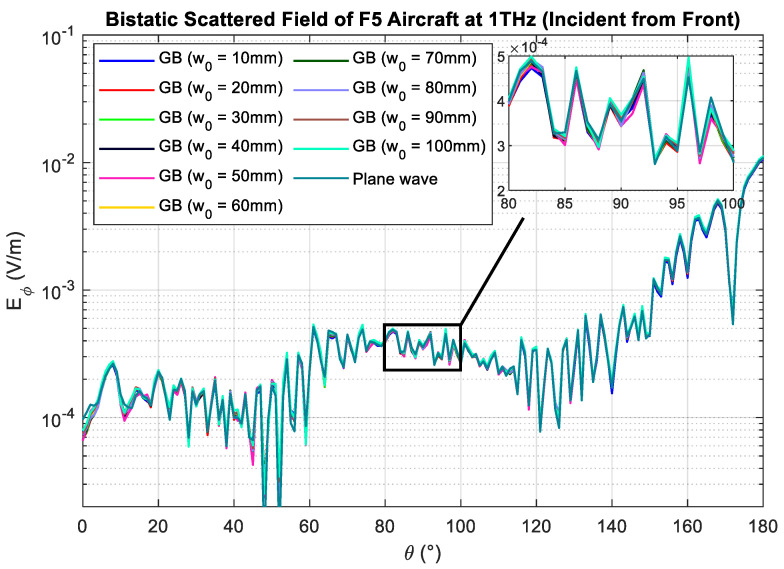
Bistatic scattered field of aircraft plotted against scattered angle. The beam is incident from the front of the aircraft (Figure 5a).

**Figure 7 sensors-23-07202-f007:**
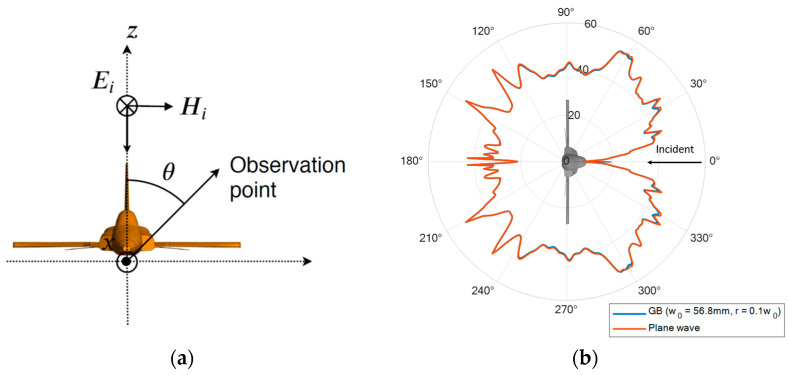
(**a**) The bistatic RCS configuration of the F5 aircraft with Gaussian beam illumination coming from the top (left). (**b**) Bistatic RCS result obtained from the predicted “optimal” beam waist.

**Figure 8 sensors-23-07202-f008:**
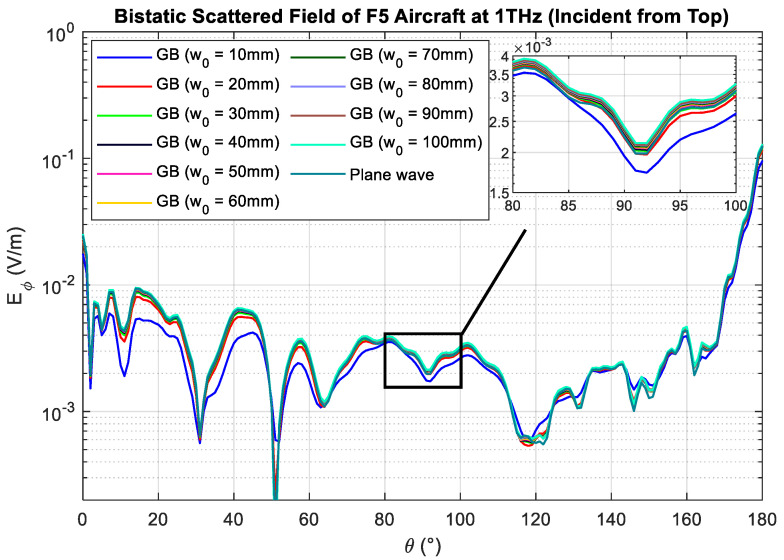
Bistatic scattered field of aircraft plotted against scattered angle. The beam is incident from the top of the aircraft (Figure 7a).

**Table 1 sensors-23-07202-t001:** Mean-squared error between $\left|E_{s,GB}\right|$ and $\left|E_{s,PW}\right|$ of the 10 mm sphere.

$w_{0}$ (mm)	MSE	$w_{0}$ (mm)	MSE
10	$7.1116\times 10^{-6}$	60	$4.8015\times 10^{-8}$
20	$4.1482\times 10^{-7}$	70	$1.6483\times 10^{-7}$
30	$5.0703\times 10^{-8}$	80	$2.3008\times 10^{-7}$
40	$1.7713\times 10^{-8}$	90	$3.2549\times 10^{-7}$
50	$3.7276\times 10^{-8}$	100	$6.8626\times 10^{-7}$

**Table 2 sensors-23-07202-t002:** Mean-squared error of RCS plotted against R (in terms of $w_{0})$.

r (mm)	MSE
${0.1w}_{0}$	$6.4834\times 10^{-7}$
$0.5w_{0}$	$3.7139\times 10^{-4}$
$1w_{0}$	0.087
${2w}_{0}$	0.2464
$5w_{0}$	12.0612

**Table 3 sensors-23-07202-t003:** Dimension of full-scaled and scaled-model F5 aircraft.

	Full-Scaled Aircraft	Scaled-Model Aircraft
Frequency	1 GHz	1 THz
*L* _1_	8.2 m	8.2 mm
*L* _2_	14.2 m	14.2 mm

**Table 4 sensors-23-07202-t004:** Mean-squared error between $\left|E_{s,GB}\right|$ and $\left|E_{s,PW}\right|$ of the scaled-model F5 aircraft. The aircraft is illuminated from the front.

$w_{0}$ (mm)	MSE	$w_{0}$ (mm)	MSE
10	$4.5499\times 10^{-10}$	60	$7.5441\times 10^{-10}$
20	$3.0406\times 10^{-10}$	70	$1.2003\times 10^{-9}$
30	$3.4263\times 10^{-10}$	80	$1.8759\times 10^{-9}$
40	$3.4263\times 10^{-10}$	90	$2.8931\times 10^{-9}$
50	$4.8520\times 10^{-10}$	100	$4.3959\times 10^{-9}$

**Table 5 sensors-23-07202-t005:** Mean-squared error between $\left|E_{s,GB}\right|$ and $\left|E_{s,PW}\right|$ of the scaled-model F5 aircraft. The aircraft is illuminated from the top.

$w_{0}$ (mm)	MSE	$w_{0}$ (mm)	MSE
10	$8.9431\times 10^{-6}$	60	$2.6626\times 10^{-8}$
20	$7.0232\times 10^{-7}$	70	$5.6800\times 10^{-8}$
30	$1.0724\times 10^{-7}$	80	$1.0227\times 10^{-7}$
40	$2.0235\times 10^{-8}$	90	$1.6673\times 10^{-7}$
50	$1.1855\times 10^{-8}$	100	$2.6029\times 10^{-7}$

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
