# Peer review of "Scaled-Model Radar Cross-Section Measurement: The Influence of the Scattered Field under Gaussian Beam Illumination—A Theoretical Analysis"

_sensors, 2023, doi:10.3390/s23167202_

Round 1

Reviewer 1 Report

This manuscript is devoted to analysis the radar cross section (RCS) obtained using Gaussian beam and plane wave illumination, which is valuable for engineers involved in designing scaled-model RCS measurements using collimated beams.

Specific comments are as follows:

1. Since the commercial software is not able to solve RCS under Gaussian beam illumination, in this manuscript, a Gaussian beam is decomposed into a sum of plane wave. How many plane waves is chosen to synthesize the Gaussian beam? It should be clearly stated.

2. This paper is focused on theoretical analysis and numerical simulations, but the title is mainly regarding with “Measurement”.

3. A section on electromagnetic scaling theory should be added.

4. In numerical simulations, the polarizations of incidence waves should be clearly stated.

Reviewer 2 Report

My comments for Authors:

1. abstract, please add obtained results for underline the novelty of paper

2. please improve the quality of figures, e.g. 3,4,6,7,8

3. discussion, please compare your solution with paper from references

Reviewer 3 Report

General evaluation:

This paper sheds light on the problem of measurement of radar cross-section (RCS) using commonly available scaled models illuminated by the Gaussian beams. It will be useful for radar and radiophysics engineers to run the scaled measurements with collimated beams. I recommend it for publication after minor revision according to the list of 5 technical comments.

Technical Comments:

1) Please change Roman numbers to Arabic everywhere in the manuscript text when you refer to the Section numbers. E.g. on page 2, line 58 please change "(see Equation (1) in Section II)." to "(see Equation (1) in Section 2)."

2) What are the criteria of an electrically quiet environment, which is suitable for the RCS measurements? Do you have any practical recommendations to do the measurements?

3) Page 4: Wrong units are mentioned in the labels displayed in Figure 1(a). Please correct "w0=40cm" to "w0=40 mm" and "wR=56.57cm" to "wR=56.57mm".

4) Page 6, line 236: Wrong figures are mentioned in the text "At different positions along z, for example, at ?=0 and ?=?0 shown in Figure 1(c) and 1(d)". Please correct the figure numbers in this text to "Figure 1(d) and 1(e)". It is clearly shown in Fig.1(e) that the beam waist is larger at the Rayleigh range ?=?0 than the waist at ?=0 in Fig.1(d).

5) Could you conclude that the bistatic RCS could be used as a radar signature independently of the object size? In other words, is it true that using the scaled models of real objects (e.g. aircraft) we could create their radar signatures and after that detect objects in the sky by their radar signatures? If it is true, such a conclusion could draw additional interest to your paper.

Round 2

Reviewer 1 Report

The revised paper addressed all my concerns regarding the first version.